# Accuracy of Modified Blue-Dye Testing in Predicting Dysphagia in Tracheotomized Critically Ill Patients

**DOI:** 10.3390/diagnostics13040616

**Published:** 2023-02-07

**Authors:** Manuel Muñoz-Garach, Olga Moreno-Romero, Rosario Ramirez-Puerta, Eugenia Yuste-Ossorio, Francisca Quintana-Luque, Manuel Muñoz-Torres, Manuel Colmenero

**Affiliations:** 1Intensive Care Medicine, University Hospital Clínico San Cecilio, 18016 Granada, Spain; 2Physical and Rehabilitation Department, University Hospital Clínico San Cecilio, 18016 Granada, Spain; 3Instituto de Investigación Biosanitaria de Granada (ibs.GRANADA), 18012 Granada, Spain; 4Endocrinology and Nutrition Division, University Hospital Clínico San Cecilio, 18016 Granada, Spain; 5Department of Medicine, University of Granada, 18016 Granada, Spain; 6Centro de Investigación Biomédica en Red Fragilidad y Envejecimiento Saludable (CIBERFES), Instituto de Salud Carlos III, 28029 Madrid, Spain

**Keywords:** dysphagia, tracheostomy, critically ill, fiberoptic endoscopic evaluation of swallowing, blue-dye test, diagnostic accuracy

## Abstract

(1) Background: Diagnosis of dysphagia in critically ill patients with a tracheostomy is important to avoid aspiration pneumonia. The objective of this study was to analyze the validity of the modified blue-dye test (MBDT) on the diagnosis of dysphagia in these patients; (2) Methods: Comparative diagnostic test accuracy study. Tracheostomized patients admitted to the Intensive Care Unit (ICU) were studied with two tests for dysphagia diagnosis: MBDT and fiberoptic endoscopic evaluation of swallowing (FEES) as the reference standard. Comparing the results of both methods, all diagnostic measures were calculated, including the area under the receiver-operating-characteristic curve (AUC); (3) Results: 41 patients, 30 males and 11 females, mean age 61 ± 13.9 years. The prevalence of dysphagia was 70.7% (29 patients) using FEES as the reference test. Using MBDT, 24 patients were diagnosed with dysphagia (80.7%). The sensitivity and specificity of the MBDT were 0.79 (CI95%: 0.60–0.92) and 0.91 (CI95%: 0.61–0.99), respectively. Positive and negative predictive values were 0.95 (CI95%: 0.77–0.99) and 0.64 (CI95%: 0.46–0.79). AUC was 0.85 (CI95%: 0.72–0.98); (4) Conclusions: MBDT should be considered for the diagnosis of dysphagia in critically ill tracheostomized patients. Caution should be taken when using it as a screening test, but its use could avoid the need for an invasive procedure.

## 1. Introduction

Approximately 25–60% of patients admitted to the intensive care unit (ICU) require mechanical ventilation and endotracheal intubation [1]. Most of these patients, especially high-risk postoperative patients, have a mechanical ventilation time of less than 3 days. However, there is a percentage of about 20% in which mechanical ventilation is prolonged [2]. When 10–14 days are exceeded, a tracheostomy is advised to avoid damage to the glottic structures and facilitate oral hygiene and patient comfort [3].

Dysphagia related to the presence of an artificial airway is a recognized complication with a variable incidence depending on several circumstances [4]. It is described in extubated patients (the so-called post-extubation dysphagia) or in patients with a tracheostomy tube who are intended to start oral feeding. This dysphagia is related to several factors, both anatomical and functional, responsible for its occurrence. Among the anatomical reasons are orotracheal intubation, continuous contact of the orotracheal tube and nasogastric tubes with pharyngo-laryngeal structures, and/or the performance of tracheostomy surgery [5]. As causes of functional alteration, the following mechanisms have been described: reduced laryngeal elevation, decreased pharyngeal sensitivity, reduced cough response, and disuse atrophy of laryngeal musculature [6]. These factors compromise the protection of the airway and reduce swallowing efficiency, thus increasing the risk of aspiration in the patient with a tracheostomy [7]. In patients with a tracheostomy tube, the presence of a cuff does not prevent the aspiration of secretions, gastric contents, and even food [8]. These aspirations are more frequent in patients with dysphagia. Therefore, the aspiration produced might lead to serious medical complications such as pneumonia, airway obstruction, and even death. The adjusted ventilator-associated pneumonia incidence density rate in Spanish ICUs is around 4 to 5 per 1000 ventilator days. [9]. Therefore, dysphagia can become a serious severe adverse event that increases the risk of mortality, risk of admission to ICU, length of hospital stay, and, consequently, increase in health care costs [10].

For the diagnosis of dysphagia associated with an artificial airway, there are two reference methods [11]. These methods are (i) Fiberoptic Endoscopic Evaluation of swallowing (FEES), which allows direct visualization of glottic structures; and (ii) VideoFluoroscopic Evaluation of Swallowing (VFES), which relies on the images obtained with X-ray equipment and an oral contrast agent. In ICU, the procedure used is FEES since it can be performed at the patient’s bedside without the need for transfer to the radiological department. As possible alternative methods, there is a thorough clinical examination [12], with the support of different procedures that attempt to show the presence of aspiration of swallowed material in the airway (substances of different volumes, consistency, and color). Among these methods is methylene blue dye, that when administered mixed with liquids or semi-solids and in an amount greater than a few drops, constitutes the so-called modified blue-dye test (MBDT) [13]. However, its validity for the diagnosis of dysphagia in patients with a tracheostomy is uncertain so far.

The purpose of this present study was to examine the internal and external validity of the modified blue-dye test (MBDT) in the diagnosis of dysphagia in critically ill patients with a tracheostomy, comparing their results with that obtained with the fiberoptic endoscopic evaluation of swallowing (FEES) as the diagnostic reference test.

## 2. Materials and Methods

### 2.1. Patients

A prospective case series of tracheostomized patients admitted to a critical care unit was conducted. The study was performed in a 22-bed ICU of a University Hospital during a 2-year period (January 2018 to February 2020). The study group was composed of all patients admitted to ICU over 18 years old with a tracheostomy and in the weaning period of ventilation. The patients were selected consecutively. All the participants were tested before being decannulated. All of them were able to maintain spontaneous ventilation for a period of 2 or more hours (or able to maintain ventilation with continuous positive pressure (CPAP < 6 cm H_2_O) with or without oxygen) and in a semi-sitting position between 60° and 90°. All the patients with neurovascular or neuromuscular diseases were excluded. None of the selected patients were taking selective serotonin reuptake inhibitors, serotonin–norepinephrine reuptake inhibitors, or monoamine oxidase inhibitors. All subjects possessed the cognitive abilities to perform the FEES and MBDT procedures and the ability to accept different textures into the mouth. All patients had a Portex tracheostomy tube with a subglottic aspiration tube.

### 2.2. Procedures

Nurses and physicians were trained in the research protocol. The physicians were board certified in critical care or rehabilitation medicine and were credentialed for FEES. Patients underwent both MBDT and FEES in the following order: the MBDT and their results were conducted and recorded by the nurse in charge of the patient. The FEES was performed according to the standard protocol reported by Langmore et al. [14]. It was carried out between 6 and 24 h after the MBDT in order to remove any methylene blue residue in the laryngeal area. Physicians responsible for the FEES procedure were blinded to the MBDT results, and the validity of MBDT was calculated using FEES as the gold standard. Patient preparation for the procedures was as follows: patients were placed in a semi-sitting position, secretions were aspirated from both the inside of the tracheostomy tube and the subglottic channel, and, finally, the tracheostomy cuffs were deflated. In this study, we used the MBDT to obtain better efficiency. The difference is that the MBDT involves the administration of food materials, such as ice, liquid, or mash, impregnated with methylene blue. We mixed 3 mL of water with 2 mL of methylene blue, obtaining 5 mL of liquid (instead of four drops of the original Evans test) [15], which corresponds to the smallest volume used in procedures testing different volumes and consistencies [12]. The 5 mL was administered in a syringe in the middle-posterior third of the tongue, and then the patient was asked to swallow. The following signs and symptoms were recorded: early/late cough, changes in voice, asphyxia, a decrease of 3 or more points in pulse oximetry saturation, and the presence of blue stains (alone or mixed with secretions or saliva) through the tracheostomy tube. If no cough or spontaneous secretions were present, suctioning into the tracheostomy tube was repeated in 15-minute intervals for an hour, and the sample obtained was examined for blue discoloration against a white background under full-room lighting. MBDT was considered positive if evidence of blue-stained material was obtained through the tracheostomy cannula (Figure 1).

The FEES studies were conducted and analyzed by three physicians. One performed FEES, and the other two watched the video monitor and checked the signs and symptoms of aspiration in the patient. A nurse gave the patient 5 mL of the water mixture with methylene blue through a syringe to the middle-posterior third of the tongue. The FEES was performed, and we introduced the laryngoscope by the nose nostril, advanced to visualize the larynx from the base of the tongue. After swallowing, we advanced to visualize the first plane of larynx and trachea (laryngeal vestibule). In this way, we directly watched all the possibilities of penetration and aspiration (Figure 2). We used the Penetration-Aspiration Scale of Rosenberg et al. [16], in which Aspiration is defined as the entry of material (secretions or liquid) into the larynx below the true vocal cords, and Penetration is when the material remains in the laryngeal vestibule, not going beyond the true vocal cords.

### 2.3. Data Analyses

After the procedures, each physician and nurse filled out a register with the results for each of the study variables. Discrepancies between the three physicians were resolved by a tie-breaker (since they were odd). Investigators were blinded to each other. Descriptive statistics were used to describe patient characteristics and endoscopic findings. The data were presented as frequencies for categorical variables and mean and standard deviation (SD) for continuous variables. The prevalence of dysphagia was estimated, considering FEES as the standard reference method. For MBDT sensitivity, specificity, negative and positive predictive values, and likelihood ratios of the test were calculated. The area under the receiver operating characteristics (ROC) curve was calculated. Results were expressed as percentages and with confidence intervals at 95%. Data were collected in a spreadsheet and analyzed with the statistical software package SSPS v.20 (SPSS, Chicago, IL, USA).

### 2.4. Ethical Considerations

The study was approved by the Ethics Committee for the Hospital. Informed consent was obtained from the patient or relatives. Data were anonymized, and no image from the patients could be used without their permission.

## 3. Results

Forty-one patients with tracheostomy participated in this study (Table 1). Eleven subjects were female, and thirty were male. The mean age was 61 years (with an interval between 28 and 82 years), and the median was 65 years. The ICU admission diagnostics were 14 patients with respiratory disease (7 pneumonia and 7 exacerbation of chronic respiratory failure), 9 patients in shock (3 hemorrhagic, 3 septic, and 3 cardiogenic), 7 patients with polytrauma, 5 patients with acute pancreatitis, 2 patients with cardiorespiratory arrest, 2 post-gastrointestinal surgery, and 2 with other disorders (status epilepticus and meningoencephalitis). According to FEES results, 29 patients gave a positive result, and, therefore, the prevalence of dysphagia in our sample was 70.7% (CI95%: 54.5–83.9). There were no apparent differences in the prevalence of dysphagia according to the reason for admission to the unit.

Table 2 shows the results for MBDT and FEES. According to FEES results (reference test), there were 23 true positives (positive results for MBDT and FEES), only 1 false positive (positive result for MBDT and negative result for FEES), 11 true negatives (negative results for both tests), and finally 6 false negatives (negative result for MBDT and a positive result for FEES). So, suctioning failed to detect the presence of blue tracheal secretions in six of the twenty-nine patients who aspirated by FEES. Thus, the MBDT showed a 20.7% false-negative error rate in aspiration detection when compared with the simultaneous FEES (CI95%: 8.0–39.7).

The results for the diagnostic performance measures are in Table 3. Sensitivity was 79.3% (CI95%: 60.3–92%), and specificity was 91.7% (CI95%: 61.5–99.8%).

The area under the curve was calculated as 0.85 ± 0.05 (CI95%: 0.72–0.98) and *p* = 0.005 (Figure 3).

## 4. Discussion

The modified methylene blue test can be used in the diagnosis of dysphagia in patients on mechanical ventilation with a tracheostomy tube due to its high positive predictive value. Its use in this context allows for a decrease in the number of instrumental diagnostic tests and, more importantly, the possibility of bronchopulmonary aspiration. However, a negative result for MBDT does not permit discarding the presence of the problem. The identification of dysphagia in a patient with a tracheostomy is important and often poses difficult and challenging clinical decisions [17]. The introduction of oral feeding in critically ill patients after a prolonged episode of mechanical ventilation is a matter between efficacy and safety [18]. Both nutritional therapy and emotional well-being may be improved with the initiation of oral intake by regaining tastes, appetite, and a subjective sense of improvement. On the other hand, pulmonary aspiration may occur, which is associated with ventilator reconnection, the development of infections, and increased mortality [19]. Due to the need for specific skills for the performance and interpretation of instrumental techniques, in most ICUs, clinical examination methods are used. The most commonly used in patients with a tracheostomy has been the MBDT, based on the original Evans blue dye test.

The MBDT has become a standard clinical tool in the evaluation of patients with tracheostomy and suspected dysphagia because it offers the advantages of economy, simplicity, and availability. Originally, it was thought that it could be used as a screening test because of its high sensitivity. It was considered more relevant to avoid pulmonary aspirations than to proceed to the initiation of oral feeding or decannulation, so it was prioritized not to have false negatives over false positives. Several studies supported the use of this test as a screening test, finding high sensitivity. O’Neill-Pirozzi et al. [20] used simultaneous videofluoroscopy and MEBD with 37 patients and reported a sensitivity of 80% and a specificity of 62%. Belafsky et al. [21] compared the expanded methylene blue test with FEES in 30 patients, only 10 of whom were mechanically ventilated, but in whom the sensitivity of the test was 100%. Winklmaier et al., in patients with head and neck cancer, the sensitivity of their MBDT protocol in predicting aspiration was 95.24%. That protocol included up to six trials and two different consistencies and amounts of test material [22].

However, several subsequent studies questioned the reliability of this procedure. Warnecke et al. [23], validating a decannulation protocol in neurological patients with tracheostomy, compared the MBDT with the FEES in 41 patients. The MBDT had a low sensitivity of 32.0% in detecting aspiration, and the negative predictive value was only 46.9%. Therefore, they concluded that the use of FEES was necessary for the decannulation and swallowing evaluation process, especially because the MBDT did not detect “silent” aspirations. Brady et al. [24], after analyzing the diagnostic performance of MBDT in relation to FEES in 21 tracheostomized patients in a rehabilitation unit, stated that it could not be used as a screening test given its low sensitivity (40%), but it could avoid or delay the use of FEES in those in whom it had detected aspiration. More recently, Fiorelli et al. [25] have studied 51 tracheotomized intensive care patients and the usefulness of MBDT in this population, with a protocol of application on 3 consecutive days. Their conclusion, finding a PPV of 100% and an NPV of 58.82%, is that it should be considered not only as an alternative to standard diagnostic invasive examinations but as a screening tool to identify which tracheostomized patients should undergo FEES or VFSS. Linhares et al. [26], studying 17 patients admitted to ICU, found that the staining test had a sensitivity of 10.0% and a specificity of 100.0% for detecting aspiration, concluding that fiberoptic endoscopic evaluation of swallowing should be used for a more comprehensive diagnosis of tracheostomized patients, especially for those at high risk of aspiration. Bechet et al. [27] published a systematic review up to 2016, but they were only able to include 6 studies, with no possibility of combining them in a meta-analysis, in which they urged further comparative studies to increase the scientific evidence to better direct clinical practice. A subsequent scoping review [28] reached the same conclusions.

Our study is in agreement with those that find a higher positive than negative predictive value and therefore question its use as a screening test for dysphagia in these patients. There are several factors that explain the lack of sensitivity (false negatives) of this test. The first factor is the lack of detection of stained secretions that remain below the cords but above the cuff (even when deflated) [29]. Several alternatives have been suggested to detect them. One would be by using the endoscope through the stoma and flexed upward, another through a subglottic channel suction (present in some types of tracheostomy cannula) for the aspiration of secretions [30], and, finally, the repetition of the test with progressively increasing amounts [31]. The second factor is the amount of aspiration (trace vs. gross). Brady et al. [24] used simultaneous videofluoroscopy and MEBD to study 20 patients. They divided their patients into two groups: those with only small amounts (trace) of aspiration and those with larger amounts of aspiration. Their results found 100% sensitivity in those with severe aspiration and 50% sensitivity in those with trace aspiration. It could be concluded from these data that MEBD is most useful in those with suspected severe aspiration. Finally, the excessive time between the two tests could lead to changes in the clinical status of dysphagia and, therefore, the results between the tests. In our study, we performed both tests in less than 24 h.

Our study has some limitations. There is great controversy about the medical use of methylene blue. In 2003, the FDA banned its use because of some reports on the occurrence of adverse effects with its administration in enteral formulations in patients with alterations in intestinal permeability (septic), as it could be absorbed into the bloodstream [32]. In Europe, such a prohibition has not taken place as the results are not considered conclusive. A recent systematic review is also in the same direction when considering it safe at the doses used for swallowing studies [33]. Alternative substances, such as food dyes or contrast agents, have been suggested, but there is little published experience to date. The clinical impact of each type of aspiration has not been defined either. If trace aspiration not detected by MBDT did not result in an increase in patient morbidity and mortality, the test would be more sensitive and could be considered a screening test, and most importantly, patients would not need an instrumental test. To avoid the confusion that could be caused by methylene blue residuals from one test on another, they were separated by 6 to 24 h. We do not believe that this short period of time could have caused different results between the tests as a consequence of a clinical change (improvement in dysphagia). Finally, although our study is one of those with the largest sample size, it is still small, and the confidence intervals of the estimators are very wide.

Based on the results of this study, our recommendation is that MBDT can be used to confirm the presence of dysphagia in tracheostomized critically ill patients, thus reducing the need to systematically perform an instrumental procedure. This may be especially important in certain situations, such as those experienced during the coronavirus pandemic, where procedures such as FEES that generated aerosols should be avoided. However, if MBDT is negative, FEES should be performed to rule out dysphagia.

## 5. Conclusions

MBDT should be considered for the diagnosis of dysphagia in critically ill tracheostomized patients. Caution should be taken when using it as a screening test, but if the result is positive, its use could avoid the need for an invasive, aerosol-generating procedure.

## Figures and Tables

**Figure 1 diagnostics-13-00616-f001:**
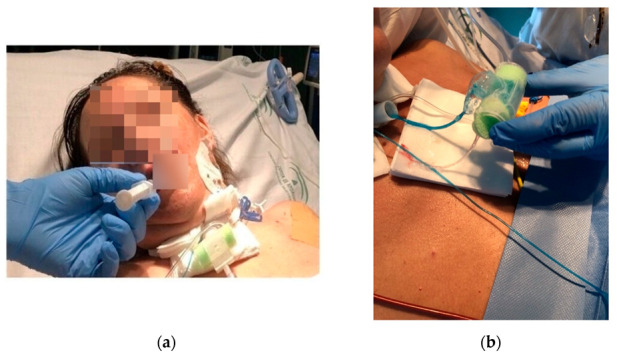
(**a**) MBDT procedure. (**b**) Positive test.

**Figure 2 diagnostics-13-00616-f002:**
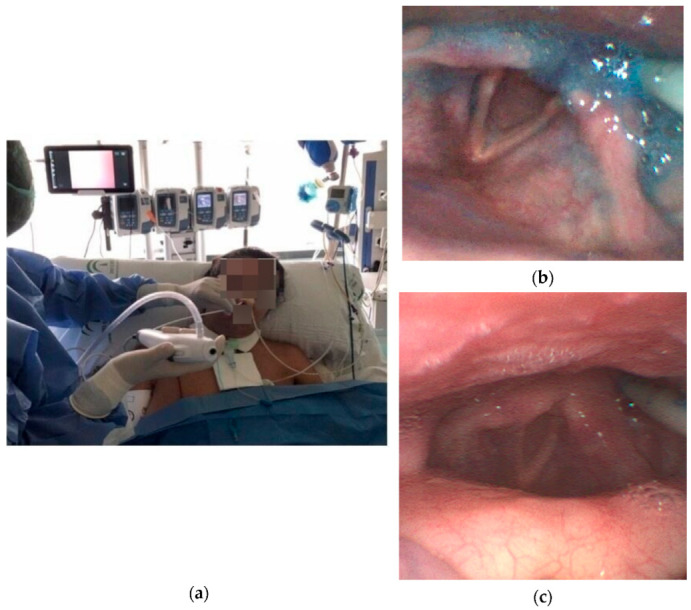
(**a**) FEES procedure. (**b**) Visualization of glottic structures. (**c**) Aspiration of methylene blue dye.

**Figure 3 diagnostics-13-00616-f003:**
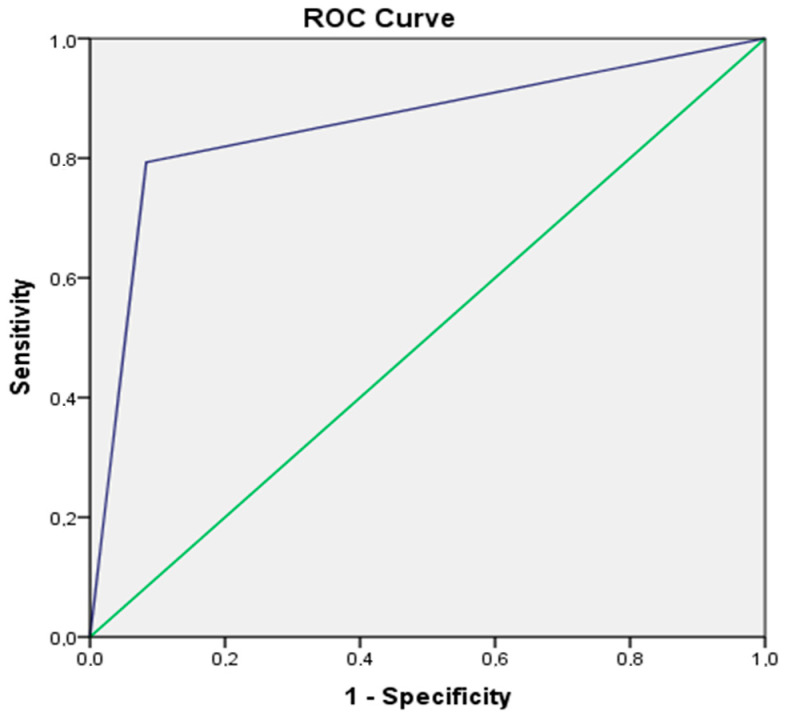
The area under the receiver operating characteristics (ROC).

**Table 1 diagnostics-13-00616-t001:** Clinical characteristics of the patient sample.

	Category	n (%), Mean (SD) or Median (IQR)
Age (years)		61 (±13.9)
Sex	Male	30 (73.2%)
	Female	11 (26.8%)
Admission diagnostics	Respiratory	14 (34.2%)
	Shock	9 (21.9%)
	Polytrauma	6 (14.6%)
	Acute pancreatitis	5 (12.2%)
	Cardiorespiratory arrest	2 (4.9%)
	Gastrointestinal surgery	2 (4.9%)
APACHE II		21.2 (±21.2)
ICU length of stay (days)		47 (33–75)
Days on mechanical ventilation		39 (27–61)
Tracheostomy days		35 (25–63)
Hospital length of stay (days)		70 (48.5–87.5)
Death rate		2 (4.9%)

**Table 2 diagnostics-13-00616-t002:** Comparison between tests.

	FEES+ for Aspiration	FEES− for Aspiration	
MBDT+	23	1	24
MBDT−	6	11	17
	29	12	41

**Table 3 diagnostics-13-00616-t003:** Diagnostic accuracy of MBDT.

Parameter	Value	CI95%
Sensitivity	79.31%	60.28% to 92.01%
Specificity	91.67%	61.52% to 99.79%
Positive Likelihood Ratio	9.52	1.44 to 62.73
Negative Likelihood Ratio	0.23	0.11 to 0.47
Disease Prevalence	70.70%	
Positive Predictive Value	95.83%	77.70% to 99.34%
Negative Predictive Value	64.74%	46.88% to 79.26%
Accuracy	82.93%	67.95% to 92.85%

## Data Availability

The data presented in this study are available on request from the corresponding author.

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
