# Peer review of "Accuracy of Modified Blue-Dye Testing in Predicting Dysphagia in Tracheotomized Critically Ill Patients"

_diagnostics, 2023, doi:10.3390/diagnostics13040616_

Round 1

Reviewer 1 Report

Comments

1.  The article  states  that the MBDI method has  only 50% sensitivity in prediction of trace aspiration. This is  weakness of the method. 

2. The small number of patients is another weakness of the article, nonetheless  it can be safely used as a complementary method.

3. Μinor spelling corrections are needed

Author Response

Thank you very much for your comments about our manuscript. 

1."The article states that the MBDI method has only 50% sensitivity in prediction of trace aspiration. This is weakness of the method".

Answer: a reference to this issue has been included in the discussion section. “ The clinical impact of each type of aspiration is also not defined. If trace aspiration not detected by MBDT did not result in an increase in patient morbidity and mortality, the test would be more sensitive and could be considered a screening test, and most importantly, patients would not need an instrumental test”

2." The small number of patients is another weakness of the article, nonetheless it can be safely used as a complementary method".

Answer: It has also been included as a limitation of the study in the discussion. “ Finally, although our study is one of those with the largest sample size, it is still small, and the confidence intervals of the estimators are very wide"

3. "Μinor spelling corrections are needed”

Answer: According to your suggestion we have rechecked the English language and style of the entire manuscript.

Reviewer 2 Report

Review report

The authors compared the diagnostic test accuracy modified blue-dye test (MBDT) with a fiberoptic endoscopic evaluation of swallowing (FEES) as the reference standard for the diagnosis of dysphagia. This was a prospective case series of tracheostomized patients, running for a two-year period  (From January  2018  to  February  2020). The principle of the test was simple: to administer blue dye-stained water (3 ml of water with 2 ml of methylene blue) and observe for the presence of blue stained  (alone or mixed with secretions or saliva)  through the tracheostomy tube. They concluded that MBDT should be considered for the diagnosis of dysphagia in critically ill tracheostomized patients. Caution should be taken when using it as a screening test, but if the result were positive,  its use could avoid the need for an invasive,  aerosol-generating endoscopic procedure.

This study addressed an important clinical conundrum of diagnosis of dysphagia in critically ill patients, especially the problem of laryngeal penetration and aspiration. The clinical significance of this study is high, and the study is relevant to clinical practice. The test is non-invasive and causes no discomfort to the patient. Further, the study has been approved by the Provincial Ethics Committee of  Granada  (protocol code  0186-N-14, 21st  January 2017).  The scientific methodology is robust and error-free. Overall, this is a high-quality research work by the authors and is praiseworthy.

Author Response

We thank to reviewer for his comments on our study.